# The Abnormal Accumulation of Lipopolysaccharide Secreted by Enriched Gram-Negative Bacteria Increases the Risk of Rotavirus Colonization in Young Adults

**DOI:** 10.3390/microorganisms11092280

**Published:** 2023-09-11

**Authors:** Yifan Wu, Shuang Pei, Jie Wu, Xinru Tu, Lingling Ren, Yanli Ji, Yuyou Yao, Yehao Liu

**Affiliations:** Department of Hygiene Inspection and Quarantine, School of Public Health, Anhui Medical University, Hefei 230000, China; wuyifan@stu.ahmu.edu.cn (Y.W.); peishuang@stu.ahmu.edu.cn (S.P.); wujie@stu.ahmu.edu.cn (J.W.); tuxinru0201@163.com (X.T.); linglren@ahmu.edu.cn (L.R.); ylji@ahmu.edu.cn (Y.J.)

**Keywords:** human rotavirus, animal-derived foods, antibiotic residues, Gram-negative bacteria, lipopolysaccharide

## Abstract

Human rotavirus (HRV) is an enteric virus that causes infantile diarrhea. However, the risk factors contributing to HRV colonization in young adults have not been thoroughly investigated. Here, we compared the differences in dietary habits and composition of gut microbiota between asymptomatic HRV-infected young adults and their healthy counterparts and investigated potential risk factors contributing to HRV colonization. Our results indicated that asymptomatic HRV-infected adults had an excessive intake of milk and dairy and high levels of veterinary antibiotics (VAs) and preferred veterinary antibiotic (PVAs) residues in urine samples. Their gut microbiota is characterized by abundant Gram-negative (G^−^) bacteria and high concentrations of lipopolysaccharide (LPS). Several opportunistic pathogens provide discriminatory power to asymptomatic, HRV-infected adults. Finally, we observed an association between HRV colonization and disrupted gut microbiota caused by the exposure to VAs and PVAs. Our study reveals the traits of disrupted gut microbiota in asymptomatic HRV-infected adults and provides a potential avenue for gut microbiota-based prevention strategies for HRV colonization.

## 1. Introduction

HRV is an enteric RNA virus that causes infantile diarrhea. It has nine species (A–I) based on its RNA sequence [1]. Among them, specie A induces over 90% of infections in human beings and results in over 200,000 deaths per year [2]. It is transmitted mainly via the feces-oral route. However, person-to-person contact has become more prominent than ever before. Since HRV can cause repeated colonization throughout life but only induce mild symptoms in adults, asymptomatic HRV-infected adults may become the source of infection [3]. However, the potential risk factors contributing to HRV colonizations require further investigation.

Many types of microbes reside in the human intestine and are known as the gut microbiota. They play important roles in host health, contributing to food digestion, intestinal barrier protection, and the synthesis of new elements [4]. However, they are vulnerable to many factors, such as diet and medication or antibiotics. It has been extensively discussed that disrupted gut microbiota is linked to chronic diseases, such as obesity and diabetes [5]. Robinson et al. reported that disrupted gut microbiota can promote the infectivity of enteric viruses [6] and the pathogenesis and replication of poliovirus by LPS through increasing viral stability and enhancing environmental fitness [7]. However, only a few cohort studies with a small number of subjects have described the characteristics of gut microbiota in HRV-infected patients [8,9]. The potential risk factors involved in the association between gut microbiota and HRV infectivity require further investigation.

Medication or antibiotic exposure is frequent because they are commonly used as growth promoters in agriculture in addition to infection treatment [10]. As a result, many types of antibiotics, especially VAs and PVAs, are deposited in the human body via food. In our previous report, we observed that the gut microbiota was disrupted by low levels of antibiotics in animal-derived foods, resulting in a high concentration of LPS in stool samples [11]. Therefore, we postulated that the disrupted gut microbiota caused by low levels of antibiotics increases the risk of HRV colonization.

The aim of this study was to compare the differences in antibiotic residues between the healthy and patient groups and to investigate the composition of the gut microbiota in the patient group. Next, we surveyed the potential risk factors for HRV infectivity and discussed the impact of exposure to medication or antibiotics on LPS biosynthesis in the gut microbiota.

## 2. Materials and Methods

### 2.1. Ethics Statement

This study was conducted in accordance with the Declaration of Helsinki’s Ethical Guidelines and approved by the Ethics Committee of Anhui Medical University. All participants were undergraduate students recruited from Anhui Medical University, Hefei City, the capital of Anhui Province, China. All the participants signed a written consent form and received a questionnaire and sample collection.

### 2.2. Collection and Process of Stool Samples

A total of 300 undergraduate students participated in the study. Sample collection, testing, and dietary investigations were conducted from November 2021 to October 2022. Fresh stool samples were first tested for HRV using an ELISA kit purchased from Shuhua Biology Co., Ltd. (Shanghai, China) and verified by real-time PCR, as reported by Kulis-Horn et al. [12]. According to the testing results, the participants with HRV were grouped into the patient group (n = 37). Simultaneously, an identical number of healthy counterparts were grouped into a healthy group (n = 37). Basic information on the participants is presented in Appendix A. All stool and urine samples were stored at −20 °C until analysis.

### 2.3. Detection of Antibiotic Residues in Urine Sample

Urine samples were collected, moved to the laboratory immediately, and stored at −20 °C. A total of 45 antibiotics from 9 categories were selected as targets for detection. Pre-treatment and subsequent analyses were performed according to our previously published method [13]. Briefly, after thawing, the urine samples were mixed with an internal standard, purified by solid-phase extraction, and analyzed by liquid chromatography-triple quadrupole tandem mass spectrometry (LC-MS/MS).

### 2.4. Dietary Investigation

A dietary investigation was conducted to determine the association between food intake and urine antibiotics levels. A semi-quantitative food-frequency questionnaire (FFQ), reported by Zhao et al. [14], was used in this study. Daily food intake, including whether the food was eaten, frequency, and amount of food at each time, was investigated according to the FFQ. All foods were classified into animal-derived and vegetable-derived foods. The questionnaire used in this study is attached as a Appendix A.

### 2.5. DNA Extraction and 16S rRNA Gene Sequencing

Approximately 100 mg of stool sample was used to extract genomic DNA using a QIAamp Fast DNA Stool Mini Kit (QIAGEN, Hilden, Germany), according to the manufacturer’s instructions. The quality of the extracted genomic DNA was assessed using a Nanodrop 2000 spectrophotometer (Thermo Scientific, Waltham, MA, USA). The V3 region of the 16S rRNA gene was amplified and the PCR product was purified. An Illumina TruSeq DNA PCR-Free Library Preparation Kit (Illumina, Hayward, CA, USA) was used to construct the sequencing libraries. Raw data were analyzed by Personalbio Technology Co., Ltd. (Shanghai, China). Qualified raw reads were processed using Quantitative Insights Into Microbial Ecology (QIIME) software (version 2.0). Operational taxonomic units (OTUs) were delineated with 97% sequence similarity. Alpha and beta diversity indices were calculated using QIIME software. Linear discriminant analysis (LEfSe) was performed using an online tool (https://huttenhower.sph.harvard.edu/galaxy accessed on 12 August 2022).

### 2.6. Metagenomic Sequencing and Following Analysis

After genomic DNA was isolated from the stool samples, metagenomic DNA libraries were constructed by Personalbio Technology Co., Ltd. (Shanghai, China). To identify the enzymes involved in LPS biosynthesis and the gut microbiota harboring enzymatic genes, we conducted a bioinformatics analysis according to a previously described method [15].

### 2.7. Determination of LPS Concentration in Stool Sample

The LPS concentration in stool samples was measured using the Bioendo Limulus Amebocyte Lysate kit (product no. EC32545, Xiamen Bioendo Company, Xiamen, China), according to the manufacturer’s instructions. Briefly, 100 mg of the stool sample was dissolved in 10 mL of purified water. After vortexing for 1 min, a clear supernatant was obtained by centrifugation at 10,000 rpm for 5 min. Standard LPS, purchased from Sigma (product no. 096M4119V, St. Louis, MO, USA), was serially diluted to construct a calibration curve (1, 0.5 EU/mL, 0.25 EU/mL 0.1 EU/mL and 0 EU/mL). The LPS concentration in stool samples was calculated according to the curve.

### 2.8. Data Statistical Analysis

Data analyses were performed using SPSS version 22.0. All data are expressed as mean ± standard deviation (SD). Statistical significance was set at *p* < 0.05. Three biological replicates were used for all experiments.

## 3. Results

### 3.1. Excessive Intake of Animal-Derived Food Is Associated with High Level of Antibiotic Residues in Urine

None of the participants had received any medication or antibiotics in the past year. However, a total of 15 types of antibiotics were detected in their urine samples (Table 1), including two HAs, four VAs, and nine PVAs. Interestingly, no difference in the concentrations of HAs was observed. The patient group had a higher concentration of 3 VAs and 6 PVAs and a lower concentration of 1 PVA than the healthy group. Among the detected antibiotics, 2 PVAs only appeared in the patient group.

As both VAs and PVAs are widely used in aquaculture and agriculture, they can enter the body via animal-derived foods, such as meat, milk, and eggs. To verify whether an excessive intake of animal-derived foods is associated with high antibiotic levels in urine, we investigated the dietary habits of all participants. Our investigation indicated that the patient group had higher percentage of animal-derived foods (53%) and lower percentage of vegetable-derived foods (47%), whereas the healthy group had a normal percentage of vegetable-derived foods (61%) and animal-derived foods (39%) (Figure 1). The above difference was attributed to the different intakes of milk and dairy, rice and wheat flour, and fresh vegetables. According to the Chinese Food Guide Pagoda (2022) released by the Chinese Nutrition Society, the recommended amount of food is 2000 g for adult per day, the amount of animal-derived foods (including milk and animal-derived foods) is 700 g, identical to 35%. Based on the information of participants’ medication history and dietary habits, our results indicated that excessive intake of milk and dairy is positively associated with high levels of antibiotic residues in the urine.

### 3.2. The Structure of Gut Microbiota in Patient Group Is Totally Different from Healthy Group

Since exposure to antibiotics can disrupt the balance of the gut microbiota, we next compared the compositional differences in the gut microbiota between the two groups. After high-throughput sequencing, qualified reads ranging from 42,591 to 45,374 were obtained. Bacterial diversity plays a crucial role in the proper functioning of the gut microbiota. To investigate whether the balance of the gut microbiota is disrupted, we first compared the within-sample diversity (α-diversity). As shown in Figure 2, the patient group had a lower Shannon index than that of the healthy group. The decreased Shannon index indicated that fewer bacterial species were recruited.

We also observed remarkable variations in the structure of the gut microbiota between the two groups. At the phylum level (Figure 3a), the patient group had a high abundance of Bacteroides (55.1%) and Proteobacteria (14.4%), whereas the healthy group had a high abundance of Firmicutes (40.6%). At the genus level (Figure 3b), the patient group had high abundances of *Bacteroides* (40.4%) and *Sutterella* (11.2%), whereas the healthy group had high abundances of *Dialister* (15.1%), *Megamonas* (14.3%), *Acidaminococcus* (14.0%), and *Alistipes* (11.2%). Next, we performed an LEfSe analysis to detect taxa that could explain the differences between the two groups. As shown in Figure 4, the dominant taxa at different taxonomic levels were determined using a linear discriminant analysis score of two or greater. We observed that *Enterococcaceae*, *Enterococcus*, *γ-proteobacteria,* and *Enterobacteriales* were enriched in the patient group, whereas *Campylobacterales*, *Brevundimonas*, *Firmicutes*, and *Lactobacillales* were enriched in the healthy group. Interestingly, although the patient group recruited less taxa than the healthy group, it was dominated by G^−^ bacteria. The above results indicate that the patient group was characterized by low bacterial diversity and a high percentage of G^−^ bacteria.

### 3.3. LPS Level in Patient Group Is Higher Than in Healthy Group

The bacterial abundance analysis showed that both Bacteroidetes and Proteobacteria have the ability to synthesize LPS [16] and were enriched in the patient group. Because exposure to medication or antibiotics can accelerate LPS release, which is involved in clinical diseases [17], we were interested in whether there was any difference in the LPS levels in stool samples between the two groups. As shown in Figure 5, the LPS level in the patient group was significantly higher, at 80.9%, than in the healthy group.

### 3.4. Several Enriched Bacteria Taxa and High LPS Level Increase the Risk of HRV Colonization

To explore the potential risk factors for HRV colonization, we first conducted a conditional logistic regression analysis to investigate the association between antibiotic exposure and the risk of HRV colonization; the results are shown in Appendix A. Although there was a significant difference in antibiotic concentration between the two groups, we did not find an association between antibiotic exposure and the risk of HRV colonization. Since exposure to antibiotics can disrupt the balance of the gut microbiota, less bacterial taxa were enriched in the patient group. Next, we conducted a conditional logistic regression analysis to investigate the association between enriched bacterial taxa and the risk of HRV colonization. As shown in Appendix A, several G^−^ bacteria showed a positive association with the risk of HRV colonization. It is also necessary to detect the role of LPS in the risk of HRV colonization because the patient group had high levels of LPS produced by G^−^ bacteria. Interestingly, the LPS level was positively associated with the risk of HRV colonization (Appendix A). These data suggest that exposure to antibiotics indirectly increases the risk of RV colonization by promoting the growth of LPS-producing bacteria.

### 3.5. Relative Abundance of Key Genes Involved in Lipid A Biosynthesis Plays an Important Role in LPS Synthesis

A previous study indicated that LPS can promote enteric viral environmental fitness by enhancing virion stability [6]. To investigate whether the imbalanced gut microbiota is beneficial to LPS synthesis, we conducted a metagenomic analysis. Fecal metagenomic sequencing data from the two groups were used to evaluate the potential of gut microbiota for LPS synthesis and to detect candidate LPS-synthesizing taxa. Because several critical enzymes, including LpxA, LpxC, LpxD, LpxH, LpxB, LpxK, WaaA, LpxL, and LpxM, are involved in LPS biosynthesis, we focused on their enrichment between the two groups based on the encoding gene sequence. As shown in Figure 6, the abundance of bacterial genes encoding LPS biosynthesis enzymes, including LpxK, LpxC, LpxL, and LpxB, was higher in the patient group than in the healthy group. Combined with the finding that stool samples in the patient group had high concentrations of LPS, these results suggest that the gut microbiota in the patient group had an elevated capacity and potential for LPS biosynthesis.

Next, we identified candidate LPS-producing bacteria based on enzymatic genes. After aligning the above enzymatic genes to the NR database, we identified the bacterial taxa in which these genes were present. Briefly, 141 genera and 512 species harbored the LpxC gene. Approximately 36 genera and 76 species harbored the LpxM gene. Approximately 90 genera and 245 species harbored WaaA. Approximately 205 genera and 674 species harbored the LpxK gene, which is the most abundant gene among the LPS-biosynthesis-related enzymatic genes. Among these LPS-synthesizing bacteria, we observed that six genera and 28 species were significantly enriched in the patient group (Figure 7). This finding was in line with the results of the LEfSe analysis, indicating that imbalanced gut microbiota in the patient group had a high potential for LPS synthesis.

## 4. Discussion

HRV is mainly transmitted via the feces-oral route [2]. However, our results indicated that 12.3% of young adults were HRV infected with no symptoms. It is possible that HRV-carrying adults are widely distributed throughout the population. Person to person dissemination also becomes an important route for HRV colonization besides the feces-oral route. Therefore, it is necessary to detect potential risk factors that can increase the risk of HRV infectivity in adults. Based on our findings, we observed that the patient group had an excessive intake of animal-derived foods, especially milk and dairy. It was positively associated with a high load of VAs and PVAs in their urine, contributing to the imbalance of gut microbiota characterized by a high abundance of G^−^ bacteria and a high concentration of LPS. Moreover, we found a positive association between HRV colonization and disruption of the gut microbiota. The results of the metagenomic analysis suggested that the imbalanced gut microbiota in the patient group had a high potential to produce LPS.

Several studies have described antibiotic residues in human urine samples [18]. Its negative effects on human health have been deeply discussed [19]. Because antibiotics are used in aquaculture and agriculture, exposure to them via daily diet is common and difficult to avoid. Therefore, more studies are required to investigate its potential negative impacts, such as its impact on the crosstalk between the gut microbiota and enteric viruses. In this study, high levels of VAs and PVAs were detected in urine samples of the patient group with no history of being on medication. After comparing the differences in dietary habits between the two groups, we observed a link between antibiotic residues and the excessive intake of animal-derived foods. Several studies have described the association between the excessive intake of animal-derived foods and antibiotic burden [10,20], reporting that the levels of VAs and PVAs in urine can reflect the intake amount of animal-derived foods because human dietary habits are not easy to change. It has been widely reported that children and young adults tend to eat more animal-derived foods, resulting in a high risk of obesity [21,22]. Our dietary investigation also found that many participants consumed excessive animal-derived foods, which was higher than the recommendation of the Chinese Nutrition Society. Our findings indicate that an excessive intake of animal-derived foods is associated with an increased risk of HRV colonization.

Gut microbiota plays an important role in maintaining human health. However, many risk factors, such as dietary habits and medication or antimicrobials, can disrupt the balance of the gut microbiota [23]. One of the markers of an imbalanced gut microbiota is its low diversity. A previous study found that HRV-infected infants have a gut microbiota with low diversity, while healthy controls have a high diversity [9]. A reduction in diversity has also been reported in children [24]. In our study, we also found decreased diversity and an imbalanced gut microbiota in the patient group. Low diversity means that some dominant taxa disappear, whereas some rare taxa become more abundant [25]. This finding was supported by LEfSe analysis, which showed that some rare taxa were abundant in the patient group. Our results suggest that a low diversity of gut microbiota is a marker of HRV carriers, regardless of age.

After comparing the composition of gut microbiota, we observed remarkable differences between the two groups. Briefly, both Bacteroides and Parabacteroides were enriched, whereas *Firmicutes* and *Euryarchaeota* were rare in the patient group. Interestingly, several findings based on HRV-induced diarrhea in infants and neonatal mice reported a decrease in the abundance of *Bacteroides* [9,26], which contradicts our findings. Since the gut microbiota in the patient group received antibiotic exposure from animal-derived foods, we deduced that Bacteroides were abundant because they are more resistant to antibiotics than other taxa, which has been reported in many studies [27,28]. Simultaneously, we noted that several probiotic genera, such as *Parabacteroides* and *Lactobacillales,* disappeared from the patient group. This decrease might be a result of exposure to antibiotics because probiotics are more vulnerable than other genera [29]. Meanwhile, we observed a significant increase in opportunistic pathogens, such as *γ-proteobacteria* and *Enterococcus*, in the patient group. Our results indicate that an imbalanced gut microbiota dominated by G^−^ bacteria is a risk factor for HRV colonization.

Many studies have reported that several metabolites secreted by gut microbiota are involved in the pathogenesis and development of human diseases [30,31]. Among these, lipopolysaccharide (LPS) has been frequently reported. Zhang et al. reported that LPS induced osteoarthritis in mice [32]. Moreover, Robinson et al. found that the gut microbiota promotes poliovirus replication and systemic pathogenesis through LPS, providing protection to enhance virion stability and promoting environmental fitness [6]. We also observed that abundant G^−^ bacteria contributed to high levels of LPS in stool samples, and most LPS-biosynthesis genes in bacteria were highly expressed in the patient group, including three key genes, LpxC, WaaA, and LpxM. These findings suggest that an imbalanced gut microbiota has an enhanced potential for LPS biosynthesis. Although there was no direct evidence describing the impact of LPS on HRV infectivity in our study, we found a positive association between high levels of LPS in stool samples and the risk of HRV colonization in young adults. Our findings are consistent with those of previous reports [6,17].

However, this study has several limitations that need to be declared. The first limitation is the small number of participants. A larger cohort is required to confirm our findings. The second limitation is the absence of data on medication or antibiotics residues in food and drinking water. Finally, although we observed a strong association between enriched bacterial taxa and the risk of HRV colonization, direct evidence describing this association is absent. An animal model is needed to explore causality.

## 5. Conclusions

In this study, we reported that the gut microbiota of young adults is disrupted by residual VAs and PVAs in food. The imbalanced gut microbiota was dominated by G^−^ bacteria, with a high potential for LPS production. Interestingly, we observed that several enriched bacterial taxa and LPS were associated with an increased risk of HRV colonization. Our findings require further clarification but they provide crucial implications for controlling HRV colonizations.

## Figures and Tables

**Figure 1 microorganisms-11-02280-f001:**
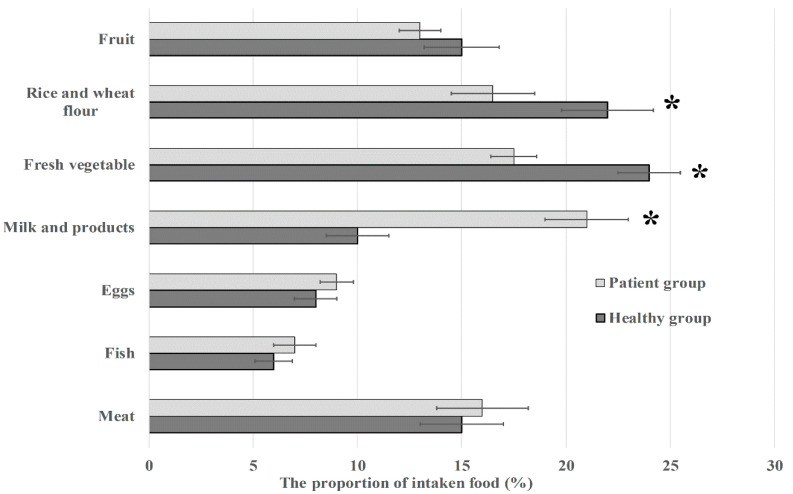
Comparison of the different consumed food types between healthy group and patient group. Meat includes pork, beef, mutton, and poultry. Fish includes fish, shrimp, and crab. Milk and products include milk, yogurt, and powdered milk. * *p* < 0.05.

**Figure 2 microorganisms-11-02280-f002:**
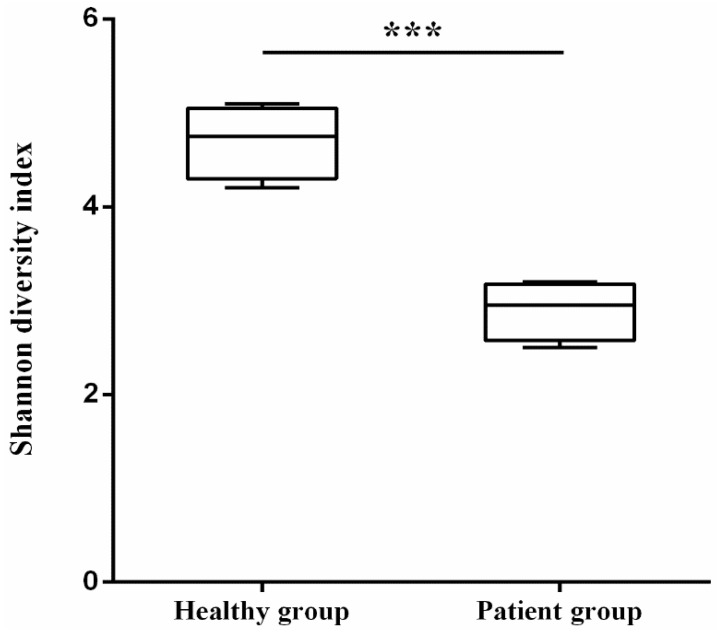
Comparison of Shannon diversity index between healthy group and patient group. The top and bottom of box represent the 75th and 25th percentile, respectively. *** *p* < 0.001.

**Figure 3 microorganisms-11-02280-f003:**
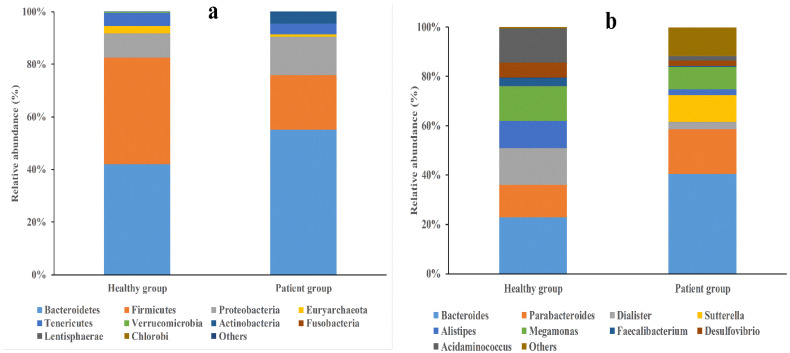
Comparison of the relative abundance of predominant bacteria at the phylum (**a**) and genus (**b**) levels.

**Figure 4 microorganisms-11-02280-f004:**
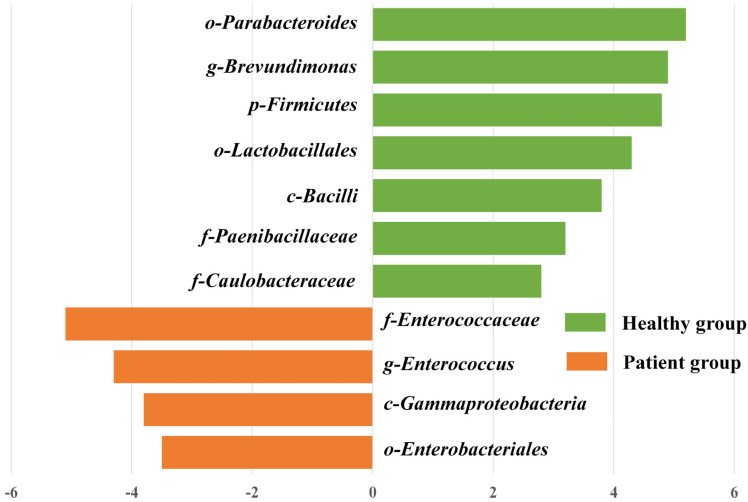
The most differentially enriched taxa between healthy group and patient group which were identified via LDA score based on the result of linear discriminant analysis effect size (LEfSe) analysis.

**Figure 5 microorganisms-11-02280-f005:**
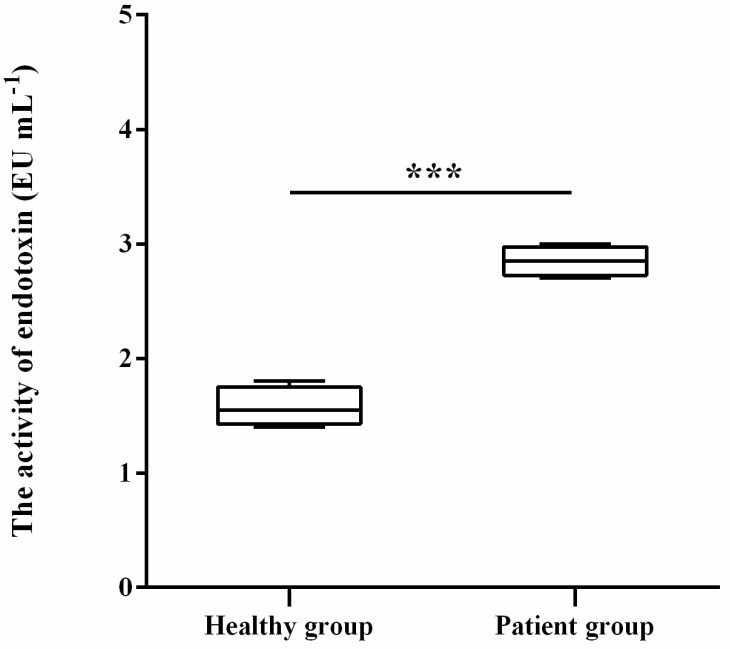
Comparison of the LPS level in stool samples between healthy group and patient group. *** *p* < 0.01.

**Figure 6 microorganisms-11-02280-f006:**
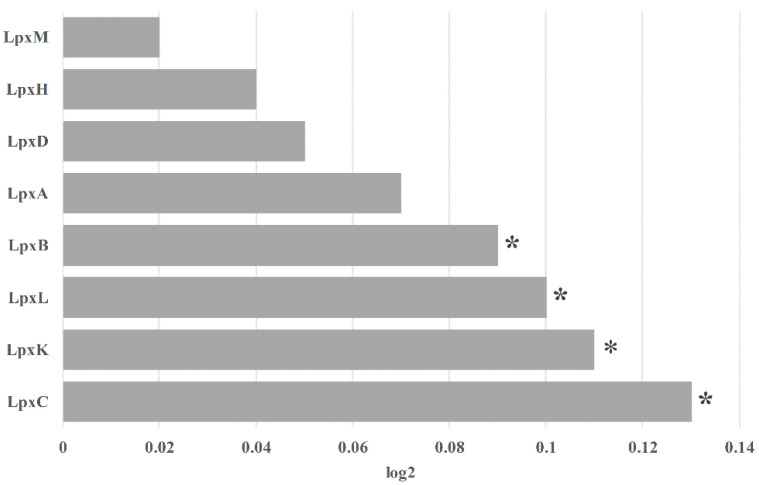
Enrichment of LPS-biosynthesis-related enzymatic genes according to the result of evaluating encoding genes using log2 (fold change of patient group/healthy group). * *p* < 0.05.

**Figure 7 microorganisms-11-02280-f007:**
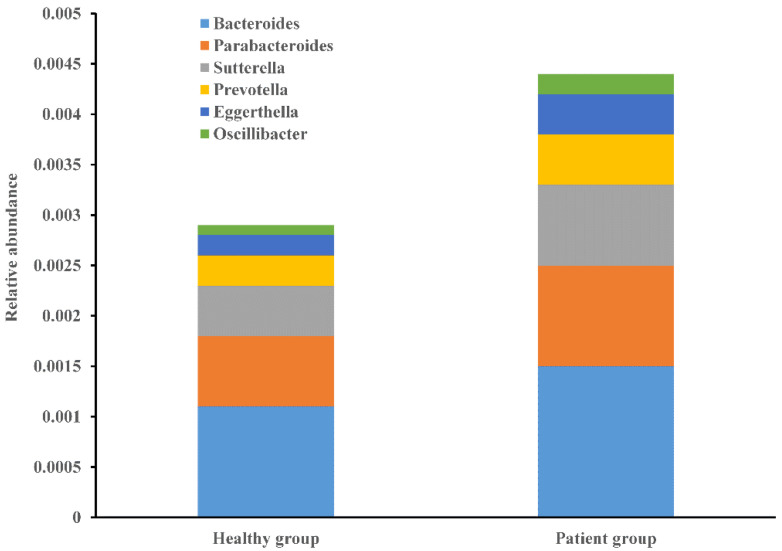
Relative abundance for the enriched genera with LPS-synthesizing potential in the two groups.

**Table 1 microorganisms-11-02280-t001:** Concentration of urine antibiotics between healthy group and patient group.

		Group	P5	P25	P50	P75	*p* Value
PVAs	Tetracycline hydrochloride	Healthy group	0.32	0.32	0.425	0.4425	0.01
		Patient group	0.6	0.6125	0.655	0.7275	
	Doxycycline hydrochloride	Healthy group	0.21	0.2375	0.325	0.3375	0.011
		Patient group	0.55	0.58	0.615	0.68	
	Oxytetracycline hydrochloride	Healthy group	0.91	0.97	1.045	1.09	0.33
		Patient group	0.86	0.875	0.955	1.065	
	Sulfametoxydiazine	Healthy group	1.07	1.1125	1.28	1.4025	0.011
		Patient group	1.5	1.53	1.66	1.77	
	Norfloxacin	Healthy group	0.8	0.83	0.9	0.95	0.025
		Patient group	0.64	0.66	0.745	0.8225	
	Sulfamethoxazole	Healthy group	0.86	0.89	1.015	1.1175	0.011
		Patient group	1.15	1.18	1.28	1.36	
	Ofloxacin	Healthy group	0.91	0.97	1.045	1.09	0.33
		Patient group	0.86	0.875	0.955	1.065	
	Penicillin V	Healthy group	0	0	0	0	0.004
		Patient group	4.55	4.6575	5.05	5.2025	
	Lomefloxacin hydrochloride	Healthy group	0	0	0	0	0.004
		Patient group	7.68	7.7975	8.155	8.445	
VAs	N4-acetylsulfamonomethoxine	Healthy group	4.22	4.2425	4.645	5.07	0.336
		Patient group	4.21	4.215	4.24	4.7375	
	Cefquinome sulfate	Healthy group	51	52.5	54.5	57	0.011
		Patient group	103	105.25	121.5	133.25	
	Ceftiofur	Healthy group	0.12	0.1275	0.175	0.2675	0.011
		Patient group	1.95	2.045	2.39	2.945	
	Cyadox	Healthy group	201	211.5	222.5	231.25	0.011
		Patient group	360	363.5	381	389.5	
HAs	Levofloxacin	Healthy group	6.12	6.1425	6.455	6.8125	0.67
		Patient group	6.22	6.2925	6.53	6.85	
	Clarithromycin	Healthy group	0.89	0.905	0.925	0.96	1
		Patient group	0.9	0.9075	0.94	0.95	

PVAs: Preferred veterinary antibiotics. VAs: veterinary antibiotics. HAs: human antibiotics. P5: 5th percentile. P25: 25th percentile. P50: 50th percentile. P75: 75th percentile. *p*-values were derived using the Wilcoxon signed-rank test.

## Data Availability

All data presented in the study are included in the article/Appendix A and further inquiries can be directed to the corresponding authors.

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
