# Peer review of "The Abnormal Accumulation of Lipopolysaccharide Secreted by Enriched Gram-Negative Bacteria Increases the Risk of Rotavirus Colonization in Young Adults"

_microorganisms, 2023, doi:10.3390/microorganisms11092280_

Round 1

Reviewer 1 Report (Previous Reviewer 1)

  • The manuscript under review focuses on investigating the factors contributing to human rotavirus (HRV) colonization in young adults, with specific attention to comparing dietary habits and gut microbiota composition between asymptomatic HRV-infected adults and their healthy counterparts. The results of this study showed that asymptomatic HRV-infected adults were found to have excessive intake of milk and dairy products, high levels of VAs and PVAs residues in urine samples, and a disrupted gut microbiota composition. The presence of several opportunistic pathogens in the gut microbiota of HRV-infected individuals was suggested to have discriminatory power. It suggested that an association between HRV colonization and the disruption of gut microbiota caused by exposure to VAs and PVAs. The study sheds light on the potential role of dietary habits, veterinary antibiotics, and gut microbiota in HRV colonization in young adults.  

  •  

The manuscript observed the associations between HRV infection and various factors such as dietary habits, gut microbiota composition, and antibiotic exposure. However, it's important to clarify whether these associations are causal or merely correlations. Establishing causation requires careful experimental design or advanced statistical techniques. 

need minor edit.

Author Response

       We sincerely thank the reviewers for their critical comments and valuable suggestions contributing to improve the quality of our manuscript. We have tried our best to address the comments from the reviewers.

       We admit it is important to clarify whether these associations are causal or merely correlations. However, this work is a case-control study, the results can’t tell us whether the association is causal correlation. To test whether the association is causal or merely correlation, in vivo and in vitro experiments were designed to detect the underlying mechanism. The part of results from our ongoing work tends to be a causal correlation, which is consistent with the references 6&7. We will report our findings in the near future.

Reviewer 2 Report (New Reviewer)

The Ms. entitled "The abnormal accumulation of lipopolysaccharide secreted by enriched gram-negative bacteria increases the risk of rotavirus colonization in young adults" is an interesting article. This article deals with the alteration of the intestinal microbiota in young people produced by antibiotics indirectly due to the presence of antibiotics found in the intake of animal products that favor the overproduction of LPS by gram-negative bacteria that in turn favor colonization by enteric viruses. The results are well founded and a satisfactory conclusion is reached. However, as you say, a larger population is necessary in the study.

Questions

1.  Why do you think that antibiotics found in food affect bacterial flora more than gram-negative bacteria?

2. How can antibiotics increase the presence of LPS?

3. How can the presence of LPS favor the colonization of enteric viruses? 

I think tat you already have an english editing in your article.

Author Response

We sincerely thank the reviewers for their critical comments and valuable suggestions contributing to improve the quality of our manuscript. We have tried our best to address the comments from the reviewers. Although our manuscript still has some flaws, we would like to receive valuable suggestions if possible. Our responses are listed below:

Point to point response:

  1. Why do you think that antibiotics found in food affect bacterial flora more than gram-negative bacteria?

Response: In our previous published paper, we observed that exposure to veterinary antibiotics via food chain disrupted gut microbiota and increased Escherichia coli virulence and drug resistance in young adults (Pathogens, 2022 Sep 18;11(9):1062. doi: 10.3390/pathogens11091062). Similar results were obtained in this work. But we did not add the finding to this work for avoiding repetition.

  1. How can antibiotics increase the presence of LPS?

Response: This phenomenon has been proved by many studies, including reference 17 in this manuscript and other reports (Clin Microbiol Rev, 2002 Jan;15(1):95-110. doi: 10.1128/CMR.15.1.95-110.2002, J Infect Dis, 1992 Jun;165(6):1033-41. doi: 10.1093/infdis/165.6.1033). As a result, we think it is not necessary to confirm this phenomenon again.

  1. How can the presence of LPS favor the colonization of enteric viruses?

Response: As supported by references 6&7, the mechanisms on the presence of LPS favor the colonization of enteric viruses have been described. Poliovirus was chosen in these studies. To detect the underlying mechanism for HRV, in vivo and in vitro experiments were designed in our ongoing work. We will report our findings in the near future.

This manuscript is a resubmission of an earlier submission. The following is a list of the peer review reports and author responses from that submission.

Round 1

Reviewer 1 Report

The manuscript under review focuses on investigating the factors contributing to human rotavirus (HRV) infection in young adults, with specific attention to comparing dietary habits and gut microbiota composition between asymptomatic HRV-infected adults and their healthy counterparts. The authors claim that asymptomatic HRV-infected individuals have distinguishing characteristics such as excessive consumption of animal-derived foods, high levels of veterinary antibiotics (VAs) and preferred as veterinary antibiotics (PVAs) residues in urine samples. The gut microbiota of these individuals was characterized by high abundance of gram-negative bacteria and lipopolysaccharide (LPS) concentrations. Furthermore, the authors found that several opportunistic pathogens were more prevalent in the gut microbiota of asymptomatic HRV-infected individuals, which provided discriminatory power for identifying this group. Finally, they observed a positive association between HRV carriage in young adults and disrupted gut microbiota caused by exposure to VAs and PVAs.

The results of this study suggest that disrupted gut microbiota may play a role in HRV infection in young adults, which provides a potential avenue for gut microbiota-based prevention strategies for HRV infection. However, it is important to note that in Figure 1, there is no significant difference in meat consumption between the HRV-infected and healthy groups. Therefore, it is more likely that the difference between these two groups is related to milk and dairy consumption rather than meat consumption. Consequently, no data support the conclusion that "excessive intake of animal-derived food is associated with high levels of antibiotic residues in urine."

In addition, there are some concerns that need to be addressed in the manuscript.

1. The manuscript would benefit from improvements in the introduction, particularly with respect to citation errors. For example, reference 3 on line 33 discusses cattle RVB infection and antibody levels, which is not directly relevant to HRV infection. Additionally, reference 5 on line 41 pertains to microbiota and autism spectrum disorder, rather than obesity or diabetes as stated in the text.

2. The term "seafood" may be used to represent fish, shrimp, and crab, while "milk and dairy" may be used instead of "milk and products."

3. More information regarding the detection of HRV is needed, including the source of the ELISA kit, reagents, and methods of the PCR assay. Additionally, reference 12 pertains to a real-time PCR protocol rather than a PCR assay, which should be clarified. The results of each detection method should be presented in the supplemental material.

4. On lines 71 to 72, “At the same time, identical quantity of healthy counterparts was grouped to health group.” The actual number should be indicated in the text.

5. The number of people in patient and healthy group need to be indicated.

5. It would be intriguing if the authors could perform a comparative analysis of microbiota diversity between HRV-infected and non-infected populations with similar dietary patterns. By controlling dietary factors, the impact of HRV infection on gut microbiota diversity could be isolated, leading to a more accurate assessment of its contribution to the infection.

6. The legend of Figure 2 requires clarification, as it appears to pertain to an index of microbiota but is not clearly described.

7. On lines 308-309, the statement "Our finding is consistent with previous reports" requires a reference to support the claim.

The manuscript's English writing is acceptable, but there are typos and minor errors that must be corrected.

Author Response

       We sincerely thank the reviewers for their critical comments and valuable suggestions contributing to improve the quality of our manuscript. We have tried our best to address the comments from the reviewers. Although our manuscript still has some flaws, we would like to receive valuable suggestions if possible. Our responses are listed below:

Point to point response:

  1. it is important to note that in Figure 1, there is no significant difference in meat consumption between the HRV-infected and healthy groups. Therefore, it is more likely that the difference between these two groups is related to milk and dairy consumption rather than meat consumption. Consequently, no data support the conclusion that "excessive intake of animal-derived food is associated with high levels of antibiotic residues in urine."

Response (R): This comment is very valuable. Milk and dairy has been used instead of animal-derived food.

  1. The manuscript would benefit from improvements in the introduction, particularly with respect to citation errors. For example, reference 3 on line 33 discusses cattle RVB infection and antibody levels, which is not directly relevant to HRV infection. Additionally, reference 5 on line 41 pertains to microbiota and autism spectrum disorder, rather than obesity or diabetes as stated in the text.

R: These two references have been replaced by proper references. Moreover, all references have been checked to avoid to similar mistake.

  1. The term "seafood" may be used to represent fish, shrimp, and crab, while "milk and dairy" may be used instead of "milk and products."

R: Hefei is an inland city. University canteen do not provide seafood for its high price, but freshwater fish, shrimp, and crab are provided. So we did not revise here to avoid misunderstanding. Milk and dairy has been used instead of milk and products.

  1. More information regarding the detection of HRV is needed, including the source of the ELISA kit, reagents, and methods of the PCR assay. Additionally, reference 12 pertains to a real-time PCR protocol rather than a PCR assay, which should be clarified. The results of each detection method should be presented in the supplemental material.

R: All information regarding ELISA kit and PCR assay have been presented in the supplemental material. We used real-time PCR here, which was described in reference 12, to detect HRV for its high sensitivity.

  1. On lines 71 to 72, “At the same time, identical quantity of healthy counterparts was grouped to health group.” The actual number should be indicated in the text.

R: It has been added.

  1. The number of people in patient and healthy group need to be indicated.

R: It has been added.

  1. It would be intriguing if the authors could perform a comparative analysis of microbiota diversity between HRV-infected and non-infected populations with similar dietary patterns. By controlling dietary factors, the impact of HRV infection on gut microbiota diversity could be isolated, leading to a more accurate assessment of its contribution to the infection.

R: This suggestion is very valuable. As our result indicated that HRV-infected individuals is rare. We only recruited 37 qualified participants in two years. It needs more time to recruit HRV-infected and non-infected populations with similar dietary patterns. We will consider this suggesting in our on-going research.

  1. The legend of Figure 2 requires clarification, as it appears to pertain to an index of microbiota but is not clearly described.

R: The legend of figure 2 has been clarified.

  1. On lines 308-309, the statement "Our finding is consistent with previous reports" requires a reference to support the claim.

R: The missing references have been added.

  1. The manuscript's English writing is acceptable, but there are typos and minor errors that must be corrected.

R: We have tried our best to correct these errors. However, it is difficult to correct them all in short time. We will keep on correcting them.

Reviewer 2 Report

The authors in this report discussed about potential risk factors that can increase the risk of HRV infections in adults. Their data revealed that that the patient group had excessive intake of animal derived foods and was related with high load of Vas and PVAs in their urine causing imbalance of gut microbiota dominated by gram negative bacteria and involvement of LPS. The study design as many limitations as already illustrated by the authors but are major concerns.

The study lacks the major co-relation between gut microbiota and its relationship with HRV infection and the clinical data is missing in the patient group which is critical to establish any pathway relationship.

Needs major revision.

Author Response

        We sincerely thank the reviewers for their critical comments and valuable suggestions contributing to improve the quality of our manuscript. We have tried our best to address the comments from the reviewers. Although our manuscript still has some flaws, we would like to receive valuable suggestions if possible.

       As declared, this study has many limitations because this study is based on cohort, and asymptomatic HRV-infected individual is rare, we cannot obtain enough data, especially the clinical data, to confirm the causality relationship between imbalanced gut microbiota and HRV infection. However, our study revealed the risk factor of HRV infection, we think it is important for controlling enteric viruses-caused infection. We have established an animal model to validate our findings and to discover the underlying mechanism.

       We have tried our best to perform language editing. However, it is difficult to make it flawless in short time. We will keep on correcting our manuscript.

Round 2

Reviewer 1 Report

The authors addressed the reviewer's comments adequately and made significant improvements, making it suitable for publication in Microorganisms.

The English writing in this manuscript is not flawless; however, the content is presented clearly and is not difficult to understand.

Reviewer 2 Report

The study still does not attract novel insights.

limitations are major concerns as stated before.